# Factorial Invariance of the Satisfaction with Life Scale (SWLS) in Mexican and Colombian University Students

**DOI:** 10.3390/bs15030277

**Published:** 2025-02-26

**Authors:** Ignacio Norambuena-Paredes, Karina Polanco-Levicán, Julio Tereucán-Angulo, José Sepúlveda-Maldonado, José Luis Gálvez-Nieto, Cristina Tavera-Cuellar, Selene Pérez-Ramírez, Crisóforo Álvarez-Violante, Roque López-Tarango

**Affiliations:** 1Department of Social Work, Universidad de La Frontera, Temuco 4780000, Chile; ignacio.norambuena@ufrontera.cl (I.N.-P.); julio.tereucan@ufrontera.cl (J.T.-A.); 2Doctoral Program in Social Sciences, Universidad de La Frontera, Temuco 4780000, Chile; 3Department of Psychology, Universidad Catolica de Temuco, Temuco 4780000, Chile; k.polanco01@ufromail.cl; 4Department of Psychology, Universidad de La Frontera, Temuco 4780000, Chile; jose.sepulveda@ufrontera.cl; 5Faculty of Psychology, Fundación Universitaria Konrad Lorenz, Bogotá 110110, Colombia; cristina.tavera@ustabuca.edu.co; 6Escuela de Estudios Superiores de Jojutla, Universidad Autónoma del Estado de Morelos, Jojutla de Juárez 62900, Morelos, Mexico; selene@uaem.mx (S.P.-R.); crisoforo.alvarezv@uaem.mx (C.Á.-V.); roque.lopezt@docentes.uaem.edu.mx (R.L.-T.)

**Keywords:** life satisfaction, subjective well-being, factorial invariance, university students, psychometric properties

## Abstract

This study aimed to evaluate the psychometric equivalence of the Satisfaction with Life Scale (SWLS) in university students from Mexico and Colombia. A non-probabilistic convenience sampling was used in five public and private universities in both countries, with a sample of 861 university students (40% men and 60% women), whose average age was 20.55 years (SD = 2.72). A six-point version of the SWLS was employed. Through confirmatory factor analysis, a unidimensional structure of the SWLS was identified in both samples, with adequate fit indices in both countries. Additionally, the factorial invariance analysis confirmed the metric and configural equivalence of the model, indicating that the factorial structure and factor loadings are comparable between both populations. The results support the use of the SWLS to assess life satisfaction in the context of university education in Mexico and Colombia.

## 1. Introduction

Life satisfaction constitutes a fundamental pillar of subjective well-being, integrating a key evaluative dimension in studies on psychology and quality of life ([9]; [31]; [36]; [41]; [44]). This construct has garnered worldwide interest due to its ability to reflect how individuals assess their lives as a whole, particularly in contexts of rapid social and cultural change ([29]; [36]; [38]). Specifically, empirical evidence on the factorial invariance of life satisfaction among university students remains limited, which is crucial for ensuring the validity and comparability of results in cross-cultural studies ([22]).

Life satisfaction is recognized as the cognitive component of subjective well-being, allowing individuals to conduct a global evaluation of their lives based on their expectations and standards ([14]; [56]; [59]; [72]). This evaluation is distinct from happiness, which includes affective dimensions such as the prevalence of positive emotions and the low occurrence of negative emotions ([4]; [25]; [37]; [39]; [55]). The distinction between the two allows life satisfaction to be considered a stable and complementary measure of the affective components of well-being ([40]; [58]).

Within studies on well-being and quality of life among university students, life satisfaction emerges as a key indicator for assessing health from a holistic perspective that goes beyond the mere absence of disease ([18]; [51]; [53]; [57]). Among this population, high life satisfaction has been associated with better academic performance, greater resilience, and effective stress management skills, fostering their overall development in both academic and professional settings ([7]; [12]; [62]).

In Mexico, research on life satisfaction encompasses different groups and contexts, such as older adults ([30]; [69]), the influence of religion ([17]), and psychometric aspects focused on aging and health ([43]). A recent study examines the invariance of the Satisfaction with Life Scale (SWLS) among adolescents from Spain and Mexico, considering gender and age differences ([24]).

In Colombia, there is a growing interest in evaluating this construct through descriptive studies ([1]; [10]; [26]; [32]; [33]; [47]) and a recent psychometric study analyzing an adolescent population ([25]). However, comparative studies on university students from Mexico and Colombia have not yet been conducted.

This research aims to contribute to this outlook by comparing the factorial invariance of life satisfaction among Colombian and Mexican university students, enhancing the understanding of well-being in student populations across different cultural contexts. The findings will provide evidence on whether the Satisfaction with Life Scale (SWLS) measures this construct equivalently in both populations, ensuring the validity and comparability of cross-cultural studies.

Additionally, these results can inform educational policies and university well-being programs by providing empirical evidence for the design of psycho-emotional support strategies in higher education ([45]; [68]). Likewise, the study will strengthen knowledge about cultural differences in the perception of life satisfaction, contributing to research in well-being psychology and future methodological adaptations for measuring subjective well-being in Latin America ([14]; [23]).

### The Satisfaction with Life Questionnaire for University Students: Description and Theoretical Structure of the Construct

The Satisfaction with Life Scale (SWLS), developed at the University of Illinois, is one of the most widely used tools for reliably measuring overall life satisfaction ([54]). Unlike other instruments that assess affective or specific dimensions of satisfaction, the SWLS provides a global and concise measure of perceived quality of life ([16]). Composed of five items, this scale allows for a quick assessment and has proven to be highly adaptable to different cultural contexts and population groups, including university students ([50]).

From a theoretical perspective, the Satisfaction with Life Scale (SWLS) is based on a unidimensional model, focusing exclusively on the cognitive evaluation of life satisfaction, without including affective components ([15]; [19]; [42]; [49]; [70]). This means that the scale measures the subjective perception of well-being based on fulfilling personal expectations rather than relying on fluctuating emotional states. This model has been widely validated, and its unifactorial structure has demonstrated consistency across multiple cross-cultural studies, facilitating international comparisons ([6]; [8]; [3]; [35]; [61]; [64], [65]).

Numerous studies have supported the factorial and convergent validity of the Satisfaction with Life Scale (SWLS) through confirmatory factor analyses, demonstrating its internal reliability and its relationship with other well-being measures ([20]; [50]; [65]). In Mexico, the SWLS has shown strong psychometric properties, correlating with economic and emotional well-being factors, and its invariance across genders and different age groups has been demonstrated, supporting its applicability in comparative studies ([24]; [43]). On the other hand, in Colombia, the scale has shown significant correlations with variables such as optimism and positive affect, reinforcing its usefulness in the study of psychological well-being ([5]; [10]; [25]; [32]).

Despite SWLS’ robust structure, cultural differences can influence the perception of life satisfaction, making a detailed analysis of its equivalence in different contexts necessary. In Mexico, social support and economic stability seem to play a more significant role ([25]; [60]), whereas in Colombia, life satisfaction is associated with optimism and mental health ([25]; [48]). These variations highlight the importance of considering the context when interpreting SWLS results since, although its factorial structure may be homogeneous, the factors influencing life satisfaction can differ between countries ([2]; [67]).

Given this context, the present study hypothesizes that the SWLS will maintain an equivalent factorial structure in the samples from Mexico and Colombia, meaning that it will retain the same number of items and the same factorial configuration in both populations. Confirming factorial invariance will allow for valid and accurate comparisons of life satisfaction between the two countries ([25]; [43]), ensuring that score differences reflect actual variations in well-being rather than measurement biases ([60]). These findings will contribute to strengthening the validity of the SWLS in cross-cultural studies in Latin America and provide key evidence for the adaptation of policies and university well-being programs in the region ([50]; [23]).

## 2. Materials and Methods

### 2.1. Procedure

For the administration of the instrument, the directors of the participating universities were contacted, and their authorization was obtained. The Ethics Committee of the University of La Frontera previously approved the study. Project File Number UFRO No. 145/23. Data were collected through an online questionnaire hosted on the QuestionPro platform. Participants were invited via emails sent by the research team, which included the informed consent form and the link to the questionnaire. This document outlined the study’s objectives, the voluntary nature of participation, the confidentiality and anonymity of the data, the absence of risks, and the right to withdraw at any time. Additionally, three judges from Colombia and Mexico reviewed the SWLS items to identify and adjust any possible colloquial expressions. Following this process, they determined that the scale would be well understood in both contexts and that no modifications were necessary, keeping the items unchanged for both countries. The data for the study were collected in both countries in June 2024.

### 2.2. Participants

The study involved the participation of 861 university students, selected through non-probabilistic sampling in five public and private universities in Mexico and Colombia. Using the QuestionPro platform, the complete responses option was selected, ensuring missing data were absent. This sample size allowed for capturing the necessary variability for multivariate psychometric analysis and provided stability in the results. Below, the characteristics of each group are described.

Mexican sample: A total of 522 Mexican students of both sexes participated (40.8% men and 59.2% women), ranging from 18 to 30 years old (M = 20.54, SD = 2.80). These students were pursuing higher education at two public universities in Cuernavaca and Mexico City, Mexico.

Colombian sample: A total of 339 Colombian students of both sexes participated (39.8% men and 60.2% women), aged 18 to 31 years (M = 20.55, SD = 2.65). These students were pursuing higher education at three private universities in Bucaramanga and Bogotá, Colombia.

To assess the equivalence of the samples between the countries, associations with the sex variable were analyzed using the chi-square test (χ^2^ (df = 2) = 0.093, *p* = 0.955), and mean age differences were examined using Student’s *t*-test (t [df = 0.859] = 0.050; *p* = 0.960). No statistically significant differences were found between the samples.

### 2.3. Instrument

In order to answer the research objectives, this study applied a cross-sectional design, with the psychometric properties of the SWLS ([16]) (see Appendix A).

To achieve this, two measurement instruments were used:

The following instruments were used to achieve the research objectives: a sociodemographic questionnaire, which collected data on participants such as age, sex, marital status, educational level, and ethnic background, among other aspects.

The Satisfaction with Life Scale (SWLS), originally developed in the United States by [16] ([16]), assesses subjective well-being by focusing on the cognitive dimension of life satisfaction through five items. Although the original version uses a 7-point scale, **this study employed a Spanish version with 6 points (1 = Strongly disagree, 6 = Strongly agree) adapted and applied to a sample of older Ecuadorians ([65]).** This adaptation addresses cultural and contextual considerations specific to the Latin American population, as in countries such as Mexico, Colombia, and Ecuador, it is common practice to shorten scales in order to optimize item comprehension and facilitate more precise responses. Moreover, the 6-point version has demonstrated robust validity and reliability in studies conducted in Ecuador and Chile, with evidence of a unidimensional structure, high internal consistency (Cronbach’s α values exceeding 0.80), and adequate convergent validity with other well-being instruments, such as the Subjective Happiness Scale (SHS), without compromising the precision in measuring life satisfaction ([64], [65]).

A study conducted in Mexico reported internal consistency with a Cronbach’s Alpha coefficient of 0.86 and McDonald’s Omega of 0.95 ([24]). In the case of Colombia, it reported a Cronbach’s Alpha coefficient of 0.89 and McDonald’s Omega of 0.87 ([60]). Other studies have demonstrated that the SWLS shows an adequate psychometric fit, confirming its unidimensional structure ([25]; [30]; [43]; [70]).

### 2.4. Data Analysis

First, descriptive analyses were conducted for each SWLS item regarding mean and standard deviation for the samples from Mexico and Colombia. Additionally, the normality of the items was verified using the Kolmogorov–Smirnov test. A confirmatory factor analysis was performed in both samples based on the established theoretical structure to explore the validity of the factorial structure of the instrument. This analysis used the MPLUS 7.11 software ([52], Los Ángeles, CA, EE.UU), employing the maximum likelihood robust (MLR) method for estimating goodness-of-fit indices. Since the data do not meet the assumption of multivariate normality, this method is appropriate for generating robust estimates, as evidenced by the results obtained through Mardia’s test for the Mexican sample (−0.9) and the Colombian sample (1.13) ([46]). Using robust maximum likelihood allows for reliable goodness-of-fit indices and precise estimates of parameters and standard errors ([27]; [28]).

To assess the goodness of fit of the models, several indices were used: the Satorra-Bentler chi-square (SB-χ^2^; [63]), the comparative fit index (CFI), the Tucker–Lewis index (TLI), and the root mean square error of approximation (RMSEA). For the CFI and TLI indices, values above 0.90 indicate an acceptable fit, while RMSEA values below 0.08 reflect a reasonable fit ([66]; [11]).

Next, the level of factorial invariance of the scale between the samples from both countries was examined through a series of factorial invariance models ([71]). The following models were included: configural invariance, which establishes the same number of factors and item structure in both groups; metric invariance, which evaluates the equivalence of factor loadings; scalar invariance, which verifies the equality of intercepts; and latent mean invariance, which compares the means of latent factors between groups. The assessment of invariance was performed based on the following criteria: ΔTLI of 0 = perfect and ≤0.01 = acceptable, ΔRMSEA ≤ 0.015, as evidence of measurement invariance ([13]).

Finally, to ensure the internal consistency of the scale, the reliability of the factors was assessed using McDonald’s Omega coefficient and standardized Cronbach’s Alpha ([21]), as well as the item-total homogeneity index.

## 3. Results

### 3.1. Descriptive and Correlational Analysis

Table 1 presents the descriptive statistics of the scale items, including means, standard deviations, skewness, and kurtosis, differentiated between the samples from Mexico and Colombia. In general, high average values are observed in both samples, but slight differences exist between countries.

In the Mexican sample, the item with the highest mean is item 4 (“So far, I have gotten the important things I want in life”) with a value of M = 4.30 and SD = 1.31, accompanied by negative skewness (−0.75) and positive kurtosis (0.20), suggesting a symmetric and slightly peaked distribution. On the other hand, the item with the lowest mean is item 5 (“If I could live my life over, I would change almost nothing”) with M = 3.72; SD = 1.64, showing a more dispersed and symmetric distribution (skewness = −0.17; kurtosis = −1.07).

In the Colombian sample, the item with the highest mean is item 2 (“The conditions of my life are excellent”) with M = 4.45 and Sd = 1.04, characterized by negative skewness (−0.73) and positive kurtosis (0.90), indicating a distribution skewed towards high values and with greater concentration around the average. On the other hand, the item with the lowest mean also corresponds to item 5 with M = 3.75; Sd = 1.54, which shows negative skewness (−0.17) and negative kurtosis (−0.92), suggesting a more dispersed and flattened distribution.

Table 2 presents the Pearson correlation coefficient (r) matrix between the scale items, where the upper diagonal corresponds to the Mexican sample and the lower diagonal to the Colombian sample. Significant and positive correlations are generally observed in both samples, although there are differences in magnitude and correlation patterns. Comparatively, the correlations in the Mexican sample tend to be higher than in the Colombian sample.

In the Mexican sample, the highest correlations are found between item 3 (“I am satisfied with my life”) and item 4 (“So far, I have gotten the important things I want in life”) with a value of r = 0.706, reflecting a strong relationship between these two dimensions of personal satisfaction. On the other hand, the lowest correlation occurs between item 1 (“In most ways my life is close to my ideal”) and item 5 (“If I could live my life over, I would change almost nothing”) with r = 0.364, indicating a moderate relationship.

In the Colombian sample, the highest correlation is found between item 3 and item 4, with r = 0.606, which also suggests a strong association between life satisfaction and the perception of having achieved important things. The lowest correlation in this sample occurs between item 2 (“The conditions of my life are excellent”) and item 5, with r = 0.257, reflecting a weaker relationship than the Mexican sample.

### 3.2. Factor Structure of the Satisfaction with Life Scale

Two separate CFAs were conducted for the Mexican and Colombian samples to analyze the factorial structure of the scale. In both cases, the goodness-of-fit indices were satisfactory. For the Mexican sample, the results were as follows: χ^2^ (df = 5) = 22.937, *p* < 0.01; RMSEA = 0.073; CFI = 0.972; TLI = 0.945; SRMR = 0.024. Similarly, the Colombian sample’s indices were also satisfactory: χ^2^ (df = 5) = 13.211, *p* < 0.01; RMSEA = 0.070; CFI = 0.975; TLI = 0.954; SRMR = 0.030. These results confirm that the model exhibits a unidimensional fit consistent with the original version (Table 3).

### 3.3. Factorial Invariance

A factorial invariance analysis was conducted to analyze the comparative results between countries. The first model evaluated was configural invariance, whose results were satisfactory [χ^2^ (df = 10) = 35.096, *p* < 0.001; CFI = 0.975; TLI = 0.951; RMSEA = 0.027], indicating that the factorial structure is equivalent between the countries. Next, the metric invariance model was evaluated by adding constraints to the factor loadings. The results showed no significant differences between the metric and configural models [χ^2^ (df = 14) = 44.737, *p* < 0.001; CFI = 0.970; TLI = 0.916; RMSEA = 0.071; ΔRMSEA = −0.005; ΔCFI = −0.005], confirming the equivalence of factor loadings between both countries. Finally, the scalar invariance model, which introduces constraints on the intercepts, was analyzed. The results indicated significant differences between the metric and scalar models. Given this outcome, modification indices (MIs) were analyzed, revealing significant differences in the intercepts of items 1, 2, 4, and 5. Consequently, a partial scalar invariance model was evaluated (Table 4), in which the constraints on these intercepts were relaxed. This model showed an improvement in fit, explaining between 30.4% and 66.4% of the item variance. However, when compared to the metric invariance model, significant differences between the two models remained.

### 3.4. Reliability Evidence

Once the unidimensional factorial structure was confirmed, the scale’s internal consistency was evaluated. The results indicate that the SWLS shows high reliability in both countries. The results show that the reliability coefficients are higher in the Mexican sample (McDonald’s ω = 0.845, Cronbach’s α = 0.839) and slightly lower in the Colombian sample (McDonald’s ω = 0.820, Cronbach’s α = 0.817).

## 4. Discussion

The objective of this article was to analyze the validity, reliability, and factorial invariance of the Satisfaction with Life Scale (SWLS) in Mexican and Colombian university students. The results show that the SWLS is a psychometrically solid instrument for use in the contexts of Mexico and Colombia and that factorial invariance reaches a plausible level in terms of configural and metric invariance.

Initially developed by [16] ([16]), the scale is a measure of subjective well-being composed of five items organized in a unidimensional structure. The empirical analyses support this one-factor structure, confirming its theoretical consistency and applicability in the evaluated contexts. In Mexico, the instrument has demonstrated consistency in studies on psychometric properties across various populations ([24]; [43]). Meanwhile, the original theoretical proposal remains consistent with recent psychometric studies analyzed in Colombia ([25]; [50]; [60]).

The confirmatory factor analysis results conducted separately in the Mexican and Colombian samples confirm the validity of the one-factor theoretical structure. The life satisfaction factor subjectively assesses how individuals perceive their well-being and quality of life in general, considering their expectations, achievements, and the balance between their aspirations and reality ([34]).

Subsequently, the level of factorial invariance of the SWLS was analyzed in both samples of university students. The first model showed configural invariance, which implies that the factorial structure of the questionnaire remains constant across both samples. This finding is consistent with previous studies that have demonstrated the configural invariance of the SWLS in different contexts and populations, suggesting that the scale measures the same construct across different cultural groups ([35]).

Regarding metric invariance, the results indicate that the factor loadings are equivalent in both samples, suggesting that the relationship between the items and the latent construct remains constant in both countries. This allows for valid comparisons of correlations and regression analyses between Mexico and Colombia. The evidence obtained in this analysis phase is consistent with previous research identifying invariance in factor loadings in cross-cultural samples ([38]).

On the other hand, the scalar invariance analysis revealed a significant deterioration in model fit when imposing the equality constraint on the intercepts. This indicates that differences in the average item values between groups may be due to variations in scale interpretation or differences in participants’ response tendencies. Similar results have been reported in previous studies, where scalar invariance is rarely fully achieved in cross-cultural comparisons, limiting the possibility of directly comparing latent means ([65]). It has been suggested that cultural factors may influence the perception of life satisfaction, which could explain this study’s lack of scalar invariance ([3]).

The reliability evidence of the scale demonstrates that the items and factors exhibit adequate internal consistency, facilitating its application in both cultural contexts ([24]).

A limitation of this study is its cross-sectional design, which prevents the analysis of changes in life satisfaction over time. Future studies could adopt a longitudinal approach to examine the evolution of this construct in university students throughout their academic trajectory and in relation to contextual factors.

Another limitation of this study lies in the composition of the sample, as the Mexican participants come from public universities, while the Colombian participants belong to private universities. Although both populations are essentially similar in terms of sociodemographic variables such as age and gender, this difference in institutional context may have influenced the results, particularly in the comparison between the two countries. Factors such as access to resources, the sociodemographic profile of students, and academic opportunities may vary depending on the type of institution, potentially affecting the interpretation of the findings. Although the sample selection was based on participant availability, future research should include greater institutional diversity to assess the stability of the results across different educational contexts and enhance the generalizability of the findings.

Future studies could incorporate broader models that include these variables, allowing for a more comprehensive analysis of the determinants of well-being in university students. This would contribute to a deeper understanding of the factors influencing life satisfaction and its stability over time.

This study validates the six-point version of the scale adapted by [65] ([65]) on Latin American population samples, specifically in Mexico and Colombia, demonstrating its robust validity and reliability in measuring subjective well-being. Consequently, it is of great interest to extend this validation to other countries in the region, which would allow for more accurate and relevant intercultural evaluations of well-being.

In summary, the findings reinforce the usefulness of the SWLS as a robust instrument for assessing life satisfaction in university contexts in Mexico and Colombia. However, identifying specific differences in item interpretation between the samples highlights the importance of considering cultural factors in future studies. This will allow for measurement adjustments and improve cross-cultural comparability, contributing to a better understanding of life satisfaction in diverse populations.

## 5. Conclusions

The findings of this study confirm the validity and reliability of the Satisfaction with Life Scale (SWLS) in university students from Mexico and Colombia. However, the lack of scalar invariance highlights the influence of cultural factors on the perception of subjective well-being, emphasizing the need to adapt measurements to ensure their comparability in cross-cultural contexts.

It is recommended for future research to expand the application of the SWLS in Latin America and incorporate the analysis of moderating variables, such as socioeconomic and educational levels, which could affect the interpretation of the scale. Additionally, it is essential to continue cross-cultural studies that assess the psychometric stability of the instrument and its sensitivity to cultural variations in the region.

Likewise, future research should further investigate the factorial invariance of the SWLS across diverse cultural contexts by employing advanced methodologies that allow for a more precise assessment of its psychometric stability. In particular, multilevel structural equation modeling is essential for identifying potential measurement biases and ensuring that the scale accurately assesses life satisfaction across different population groups. Implementing such approaches would contribute to the development of more robust instruments, enabling adjustments that better reflect the sociocultural specificities of each context and enhancing the cross-cultural validity of subjective well-being measurements.

In this regard, the present study provides valuable empirical evidence supporting the psychometric soundness of the SWLS among university students in Mexico and Colombia, reinforcing its utility in subjective well-being research. However, the identification of differences in item interpretation highlights the need for a context-sensitive approach to assessing life satisfaction. Addressing these variations will facilitate the development of measurement instruments that are more responsive to cultural diversity, improving the accuracy of cross-cultural studies and contributing to a deeper understanding of well-being among university populations in Latin America.

## Figures and Tables

**Table 1 behavsci-15-00277-t001:** Descriptive statistics, compared between countries.

Items	Mean	Std. Dev.	Skewness	Kurtosis
1 In many ways, your life is close to ideal.	3.89	1.15	−0.48	0.16
2 Your living conditions are excellent.	4.09	1.22	−0.68	0.20
3 You are satisfied with your life.	4.27	1.30	−0.79	0.21
4 So far, you have gotten the important things you have wanted in life.	4.34	1.27	−0.78	0.34
5 If you could live your life over, you wouldn’t change anything.	3.73	1.60	−0.17	−1.01
Mexican Sample				
1 In many ways, your life is close to ideal.	3.86	1.20	−0.56	0.17
2 Your living conditions are excellent.	3.85	1.28	−0.55	−0.18
3 You are satisfied with your life.	4.16	1.35	−0.72	−0.02
4 So far, you have gotten the important things you have wanted in life.	4.30	1.31	−0.75	0.20
5 If you could live your life over, you wouldn’t change anything.	3.72	1.64	−0.17	−1.07
Colombian Sample				
1 In many ways, your life is close to ideal.	3.94	1.08	−0.30	−0.02
2 Your living conditions are excellent.	4.45	1.04	−0.73	0.90
3 You are satisfied with your life.	4.43	1.21	−0.86	0.61
4 So far, you have gotten the important things you have wanted in life.	4.40	1.20	−0.82	0.56
5 If you could live your life over, you wouldn’t change anything.	3.75	1.54	−0.17	−0.92

K-S test = Kolmogorov–Smirnov Test.

**Table 2 behavsci-15-00277-t002:** Pearson’s r correlation matrix: Upper diagonal—Mexican sample; lower diagonal—Colombian sample.

Items	1 In Many Ways, Your Life Is Close to Ideal.	2 Your Living Conditions Are Excellent.	3 You Are Satisfied with Your Life.	4 So Far, You Have Gotten the Important Things You Have Wanted in Life.	5 If You Could Live Your Life Over, You Wouldn’t Change Anything.
1 In many ways, your life is close to ideal.	1	0.575 **	0.517 **	0.482 **	0.364 **
2 Your living conditions are excellent.	0.479 **	1	0.662 **	0.590 **	0.365 **
3 You are satisfied with your life.	0.496 **	0.562 **	1	0.706 **	0.429 **
4 So far, you have gotten the important things you have wanted in life.	0.526 **	0.482 **	0.606 **	1	0.420 **
5 If you could live your life over, you wouldn’t change anything.	0.390 **	0.257 **	0.443 **	0.469 **	1

** The correlation is significant at the 0.01 level (two-tailed).

**Table 3 behavsci-15-00277-t003:** Factor structure of the Satisfaction with Life Scale.

Ítems	Factor Loadings	S.E.	Est./S.E.	Two-Tailed *p*-Value
Mexico				
It1	0.642	0.040	16.245	*p* < 0.01
It2	0.774	0.026	29.326	*p* < 0.01
It3	0.859	0.030	28.497	*p* < 0.01
It4	0.796	0.026	30.636	*p* < 0.01
It5	0.508	0.042	11.979	*p* < 0.01
Colombia				
It1	0.671	0.040	16.630	*p* < 0.01
It2	0.657	0.044	14.974	*p* < 0.01
It3	0.791	0.040	19.990	*p* < 0.01
It4	0.773	0.035	21.977	*p* < 0.01
It5	0.551	0.051	10.805	*p* < 0.01

**Table 4 behavsci-15-00277-t004:** Factorial invariance between Mexico and Colombia.

Model	χ^2^ (df)	RMSEA	CFI	TLI	SRMR	ΔRMSEA	ΔCFI	Decision
Configural invariance	35.096 (10)	0.076	0.975	0.951	0.027			Accepted
Metric invariance	44.737 (14)	0.071	0.970	0.957	0.055	−0.005	−0.005	Accepted
Scalar invariance	95.003 (18)	0.100	0.924	0.916	0.100	0.029	−0.046	Rejected
Partial scalar invariance	1289.9 (20)	0.062	0.984	0.974	0.040	−0.009	0.013	Rejected

Note: χ^2^, chi-squared; df, degrees of freedom; CFI, comparative fit index; TLI, Tucker–Lewis index, RMSEA, root mean square error of approximation; SRMR, standardized root mean square residual.

## Data Availability

The dataset for the study is available from the corresponding author upon reasonable request due to ethical restrictions.

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
