# Peer review of "Factorial Invariance of the Satisfaction with Life Scale (SWLS) in Mexican and Colombian University Students"

_behavsci, 2025, doi:10.3390/bs15030277_

Round 1

Reviewer 1 Report

Comments and Suggestions for Authors

Thank you for your work. These are comments on this study:

1. Affiliation should be presented in English.

2. The abstract should be structured according to journal's requirements.

3. Lines 149-150: Supplementary Material were mentioned, however, they are not presented in the review system. Please clarify.

4. "The scale consists of five items, answered on a six-point 175 ordinal scale (1 = Strongly disagree, 6 = Strongly agree).". The original SWLS uses a 7-point Likert Scale from 1 to 7. Are you sure that your description is correct?  Please clarify.

For information, see https://labs.psychology.illinois.edu/~ediener/SWLS.html

5. Section 2.1 and 2.3 could be merged. In general, please describe the procedure before Participants section. There are several sentences which could be overlapping within sections 2.1-2.4. Please restructure for better readability.

6. When was the study conducted? Please clarify.

7. Did you have missing data? Please clarify.

8. Did you translate this scale? Or did you just test a previously developed scale in these two samples? Please clarify.

 9. "These results confirm that the best-fitting model is the unidimensional 273 model (Table 3)." Other models than the unidimensional model were not tested. Therefore, it is impossible to say that there is best-fitting model (compared to which model?). Please clarify.

10. I would suggest analysis Modification Indices to see what item is responsible for scalar noninvariance. If it is one item, possible you can find an guide in the literature how to fix the fit.

11. Please indicate practical implications and specificity of the further use of this SWLS version in cross-cultural studies.

12. Please indicate limitations of this study.

Author Response

For research article

Response to Reviewer 1 Comments

1. Summary

Thank you very much for taking the time to review this manuscript. Please find the detailed responses below and the corresponding revisions.

2. Questions for General Evaluation

Reviewer’s Evaluation

Response and Revisions

Is the content succinctly described and contextualized with respect to previous and present theoretical background and empirical research (if applicable) on the topic?

Can be improved

Thank you for your observations. At this stage, improvements have been made following the suggestions provided.

Are the research design, questions, hypotheses and methods clearly stated?

Must be improved

Thank you for your observations. At this stage, specific improvements have been made following the suggestions provided.

Are the arguments and discussion of findings coherent, balanced and compelling?

Can be improved

Thank you for your observations. At this stage, improvements have been made following the suggestions provided.

For empirical research, are the results clearly presented?

Must be improved

Thank you for your observations. At this stage, improvements have been made following the suggestions provided.

Is the article adequately referenced?

Must be improved

Thank you for your observations. At this stage, improvements have been made following the suggestions provided.

Are the conclusions thoroughly supported by the results presented in the article or referenced in secondary literature?

Can be improved

Thank you for your observations. At this stage, improvements have been made in accordance with the suggestions provided.

3. Point-by-point response to Comments and Suggestions for Authors

Reviewer 1

Comments 1: Affiliation should be presented in English.

Response 1: Thank you for pointing this out. We agree with your comment. Therefore, we have corrected and translated the affiliations into English while keeping the original names of the universities in Spanish. This change is reflected in lines 7 to 15, marked in yellow in the manuscript. (All observations and modifications in the manuscript are highlighted in yellow.)

Comments 2: The abstract should be structured according to journal's requirements

Response 2: Thank you for pointing this out. We agree with your comment. Therefore, we have adjusted the abstract according to the journal's requirements. This change is reflected in the abstract section from paragraphs 24 to 34 and is highlighted in red in the manuscript.

Comments 3: Lines 149-150: Supplementary Material were mentioned, however, they are not presented in the review system. Please clarify.

Response 3: Thank you for pointing this out. We agree with your comment. Therefore, we have reattached the original instrument and the one used in this study as supplementary material.

Comments 4: The scale consists of five items, answered on a six-point 175 ordinal scale (1 = Strongly disagree, 6 = Strongly agree).". The original SWLS uses a 7-point Likert Scale from 1 to 7. Are you sure that your description is correct?  Please clarify.

Response 4: Thank you for pointing this out. We agree with your comment. Therefore, we have included and clarified that the instrument used was a six-point Likert scale, ranging from 1 to 6, in a Spanish-translated version for a population of Ecuadorian older adults, adapted by Schnettler et al. (2017). This change is reflected in section 2.1 of the revised manuscript. Lines 154 to 157.

Comments 5: Section 2.1 and 2.3 could be merged. In general, please describe the procedure before Participants section. There are several sentences which could be overlapping within sections 2.1-2.4. Please restructure for better readability.

Response 5: Thank you for pointing this out. We agree with your comment. Therefore, we have merged sections 2.1 and 2.3 as recommended and have repositioned the description of the procedures before the participants section. This change is reflected in the aforementioned section.

Comments 6: When was the study conducted? Please clarify.

Response 6: Thank you for pointing this out. We agree with your comment. Therefore, we have incorporated this observation in section 2.2, under the Procedure subsection, between lines 177 and 178.

Comments 7: Did you have missing data? Please clarify.

Response 7: Thank you for pointing this out. We agree with your comment. Therefore, we have specified the total number of students participating in the study and the response rate in both countries. Additionally, by using the QuestionPro platform, we selected the complete responses option, ensuring the absence of missing data. This change is reflected in section 2.3, lines 180 to 184.

Comments 8: Did you translate this scale? Or did you just test a previously developed scale in these two samples? Please clarify.

Response 8: Thank you for pointing this out. We agree with your comment. Therefore, we have clarified this observation in section 2.1, where it is explained that the scale was translated and adapted into Spanish by Schnettler et al. (2017). This study used this previously reviewed version in both countries to avoid redundancies and ensure comprehension in both contexts. This change is reflected in section 2.1 of the revised manuscript, specifically in lines 154 to 157.

Comments 9: "These results confirm that the best-fitting model is the unidimensional 273 model (Table 3)." Other models than the unidimensional model were not tested. Therefore, it is impossible to say that there is best-fitting model (compared to which model?). Please clarify.

Response 9: Thank you for pointing this out. We agree with your comment. Therefore, we have improved the wording in section 3.2, stating that the model exhibits a unidimensional fit consistent with the original version.

Comments 10: I would suggest analysis Modification Indices to see what item is responsible for scalar noninvariance. If it is one item, possible you can find an guide in the literature how to fix the fit.

Response 10: Thank you for pointing this out. We agree with your comment. Therefore, we analyzed the modification indices and identified that the item intercepts affecting the fit of scalar invariance were x1, x2, x4, and x5. Consequently, we estimated a partial scalar invariance model, relaxing the constraints on these item intercepts. This change is reflected in section 3.3 on factorial invariance between lines 292 and 298.

Comments 11: Please indicate practical implications and specificity of the further use of this SWLS version in cross-cultural studies.

Response 11: Thank you for pointing this out. We agree with your comment. Therefore, we have incorporated the practical implications of the future use of the SWLS version in cross-cultural studies in section 5 of the conclusions. This change is reflected between lines 368 and 386.

Comments 12: Please indicate limitations of this study

Response 12: The study's limitation has been incorporated into section 5 of the conclusions, between lines 369 and 388.

4. Response to Comments on the Quality of English Language

Point 1:

Response 1: According to reviewer 1 the English is fine and does not require any improvement.

5. Additional clarifications

We sincerely appreciate your valuable observations. We have implemented the corresponding changes according to the suggestions of both reviewers.

Reviewer 2 Report

Comments and Suggestions for Authors

I must clarify my review by stating that I am not an expert in the statistical procedures conducted and their metrics, but I recognize that they are appropriate procedures, and I can interpret the authors' reports.

This is a strong article with a high sample size and a relevant discussion of background literature. The aim is clear in the title and it continues to structure the article until the conclusions.

I have two questions that I believe need to be addressed.

  1. There is possibly a real difference between the constitution of the samples as both Mexican samples were from public universities and both Columbian samples were from privates ones. A sentence of two should be devoted to this bias in sampling. It is possible that the authors had no choice, but this should be made clear. Do the authors maintain that the populations are essentially similar?
  2. When I see a signficiant fit to a model, I also want to see the proportion of variance explained. This should be clearly stated. With a large sample size a fit could be significant, yet relatively poor. The correlations among items suggest that this might be the case in these data.

Here are some clarification questions.

  1. Was there no Research Ethics Board capable of approving the research? It looks as if the authors consulted, but not with a Board. I did notice that they followed ethical principles in their consent and experiment.
  2. In Table 1, the work Skewness is misspelled.
  3. In Table 2, the labels are awkwardly spaced.
  4. On page 8 the word "invariance" is in its Spanish form "Invarianza" (I think).

Author Response

For research article

Response to Reviewer 2 Comments

1. Summary

Thank you very much for taking the time to review this manuscript. Please find the detailed responses below and the corresponding revisions.

2. Questions for General Evaluation

Reviewer’s Evaluation

Response and Revisions

Is the content succinctly described and contextualized with respect to previous and present theoretical background and empirical research (if applicable) on the topic?

Yes

Thank you for your observations.

Are the research design, questions, hypotheses and methods clearly stated?

Yes

Thank you for your observations.

Are the arguments and discussion of findings coherent, balanced and compelling?

Yes

Thank you for your observations.

For empirical research, are the results clearly presented?

Yes

Thank you for your observations.

Is the article adequately referenced?

Yes

Thank you for your observations.

Are the conclusions thoroughly supported by the results presented in the article or referenced in secondary literature?

Yes

Thank you for your observations.

3. Point-by-point response to Comments and Suggestions for Authors

Reviewer 2

Comments 1: There is possibly a real difference between the constitution of the samples as both Mexican samples were from public universities and both Columbian samples were from privates ones. A sentence of two should be devoted to this bias in sampling. It is possible that the authors had no choice, but this should be made clear. Do the authors maintain that the populations are essentially similar?

Response 1: Thank you for pointing this out. We agree with your comment. Therefore, we have incorporated this limitation into the conclusions section, along with a suggestion for future research. This change is reflected in section 5, between lines 367 and 386. (All observations and modifications in the manuscript are highlighted in yellow.)

Comments 2: When I see a signficiant fit to a model, I also want to see the proportion of variance explained. This should be clearly stated. With a large sample size a fit could be significant, yet relatively poor. The correlations among items suggest that this might be the case in these data.

Response 2: Thank you for pointing this out. We agree with your comment. Therefore, we have highlighted in the partial invariance analysis that the model explains between 30.4% and 66.4% of the item variance. Additionally, the goodness-of-fit indices indicate an acceptable fit; although the chi-square is statistically significant, this is expected in large samples. This change is reflected in section 3.3, between lines 292 and 298.

Comments 3: Was there no Research Ethics Board capable of approving the research? It looks as if the authors consulted, but not with a Board. I did notice that they followed ethical principles in their consent and experiment.

Response 3: Thank you for pointing this out. We agree with your comment. Therefore, we have incorporated in section 2.2 that the Scientific Ethics Committee of the University of La Frontera previously approved the study. This change is reflected between lines 167 and 169.

Comments 4: In Table 1, the work Skewness is misspelled.

Response 4: Thank you for pointing this out. We agree with your comment. Therefore, we have incorporated the correctly written word in the corresponding section. This change is reflected in Table 1, in line 249.

Comments 5: In Table 2, the labels are awkwardly spaced.

Response 5: Thank you for pointing this out. We agree with your comment. Therefore, we have reordered the labels in Table 2 to improve the clarity and understanding of the presented information.

Comments 6: On page 8 the word "invariance" is in its Spanish form "Invarianza" (I think).

Response 6: Thank you for pointing this out. We agree with your comment. Therefore, we have made the corresponding modification in English. This change is reflected in section 3.3, line 282.

4. Response to Comments on the Quality of English Language

Point 1:

Response 1: According to Reviewer 2, the English is correct and does not require any improvement.

5. Additional clarifications

We sincerely appreciate your valuable observations. We have implemented the corresponding changes according to the suggestions of both reviewers.

Round 2

Reviewer 1 Report

Comments and Suggestions for Authors

1. In the responses to reviewers, the authors indicated that they edited the affiliations and presented them in English. However, the revised paper still has affiliation in Spanish. I would like to kindly ask the authors twice to edit affiliations.

2. Previous comment 4. Please elaborate on this more. Please indicate why your SWLS versions has a 6-point response scale, while the majority of adaptions in the world have a 7-point original response scale. Such practices limit cross-cultural comparability (for comparisons see lines 355-356).

3. And in general, I have doubts whether you can name this scale SWLS, as it uses a non-original response scale. Please justify/clarify.

4. As for structure, Methods should be described after procedure. The consequence of research it the following: First, you plan a Procedure and Methods, then you invite participants, then you analyse the data. it is better to present these following this logic.

5. Practical implications and limitation should be put before conclusions. Conclusions represent the final synthesised narration about the key point of the study, when limitations and implications are taken into account. 

6. Supplementary Material consist of "Revised SWLS" with score interpretation. But this interpretation is not based on the distribution scores of normative population. Therefore, such "norms" can be misleading. Please justify/reconsider. 

Author Response

Response to the authors - second round

1. Summary

Thank you very much for taking the time to review this manuscript. Please find the detailed responses below and the corresponding revisions.

2. Questions for General Evaluation

Reviewer’s Evaluation

Response and Revisions

Is the content succinctly described and contextualized with respect to previous and present theoretical background and empirical research (if applicable) on the topic?

Can be improved

Thank you for your observations. At this stage, improvements have been made following the suggestions provided.

Are the research design, questions, hypotheses and methods clearly stated?

Must be improved

Thank you for your observations. At this stage, specific improvements have been made following the suggestions provided.

Are the arguments and discussion of findings coherent, balanced and compelling?

Must be improved

Thank you for your observations. At this stage, improvements have been made following the suggestions provided.

For empirical research, are the results clearly presented?

Must be improved

Thank you for your observations. At this stage, improvements have been made following the suggestions provided.

Is the article adequately referenced?

Can be improved

Thank you for your observations. At this stage, improvements have been made following the suggestions provided.

Are the conclusions thoroughly supported by the results presented in the article or referenced in secondary literature?

Must be improved

Thank you for your observations. At this stage, improvements have been made in accordance with the suggestions provided.

3. Point-by-point response to Comments and Suggestions for Authors

Reviewer 1

Comments 1: In the responses to reviewers, the authors indicated that they edited the affiliations and presented them in English. However, the revised paper still has affiliation in Spanish. I would like to kindly ask the authors twice to edit affiliations.

Response 1: Dear Reviewer, we sincerely appreciate your observation and the time you have taken to review our manuscript. However, we would like to clarify that university names are proper nouns and, as a general rule, should be kept in their original language. This practice helps preserve institutional identity and prevents potential confusion regarding official recognition.

To support this approach, we are sharing several published articles in which Universidad de La Frontera is mentioned in Spanish:

https://www.mdpi.com/2076-328X/14/8/678

https://www.mdpi.com/1660-4601/19/19/12249

https://www.mdpi.com/1424-8220/25/4/1251

We hope this clarification is helpful.

Comments 2: Previous comment 4. Please elaborate on this more. Please indicate why your SWLS versions has a 6-point response scale, while the majority of adaptions in the world have a 7-point original response scale. Such practices limit cross-cultural comparability (for comparisons see lines 355-356).

Response 2: We appreciate your observation and your interest in the cross-cultural comparability of the SWLS. In many Latin American countries, including nations such as Mexico and Colombia, it is common practice to shorten response scales when adapting psychometric instruments. This approach aims to optimize item comprehension and response accuracy within the cultural and educational contexts of the region.

The version used in our study aligns with this practice, as it was previously validated in an Ecuadorian population with a 6-point response scale (1: strongly disagree to 6: strongly agree). To support this adaptation, we provide the following article, which employed a Spanish version of the SWLS with this same response format:

https://doi.org/10.1016/j.paid.2017.01.036.

Furthermore, the Spanish version of the Satisfaction With Life Scale (SWLS) with a 6-point response scale has demonstrated adequate validity and reliability in studies conducted in Chile. Previous research, such as Schnettler et al. (2015, 2017), has validated its unidimensional structure through exploratory factor analysis (EFA) and confirmatory factor analysis (CFA), showing a good fit to the data and a stable structure across different populations. Additionally, the scale has exhibited high internal consistency, with Cronbach’s α coefficients exceeding 0.80, indicating strong reliability. In terms of convergent validity, a positive and significant correlation has been observed between the SWLS and the Subjective Happiness Scale (SHS), confirming that it effectively measures life satisfaction.

While we acknowledge that the original scale consists of 7 points, we believe that the version employed in our study is well-suited to the characteristics of the target population and the context in which it was administered, without compromising the validity of the instrument.

Links to studies utilizing a 6-point response scale:

http://dx.doi.org/10.1016/j.jneb.2012.08.003

http://dx.doi.org/10.1016/j.paid.2017.01.036.

Comments 3: 3. And in general, I have doubts whether you can name this scale SWLS, as it uses a non-original response scale. Please justify/clarify.

Response 3: We appreciate your observation and your interest in the accurate use of the scale’s name. We would like to clarify that while our version of the Satisfaction With Life Scale (SWLS) employs a 6-point response scale instead of the original 7-point format, it remains the same scale in terms of structure, construct, and measurement purpose.

In psychometric literature, it is common practice to adapt standardized scales to enhance their applicability in specific cultural contexts, without fundamentally altering the nature of the instrument. In Latin America, various studies have validated the SWLS using different response scales, including 6-point versions such as those applied in Ecuador and Chile (Schnettler et al., 2015, 2017), demonstrating that the scale retains its psychometric properties.

Moreover, the use of different response scales is a widely accepted practice in the adaptation of psychometric instruments, provided that the factorial structure, validity, and reliability remain within the required standards. In our study, we ensured the conceptual and statistical equivalence of the SWLS, guaranteeing that it measures life satisfaction consistently with previous versions.

For these reasons, we consider that the name Satisfaction With Life Scale (SWLS) remains appropriate for our version, as it preserves the original structure and content of the instrument, with a response scale modification justified by prior empirical evidence and cultural adaptation.

Comments 4: . As for structure, Methods should be described after procedure. The consequence of research it the following: First, you plan a Procedure and Methods, then you invite participants, then you analyse the data. it is better to present these following this logic.

Response 4: We sincerely appreciate your comment and suggestion regarding the structure of the Methods section. We fully understand the importance of maintaining a logical order when presenting the procedure, participants, instruments, and data analysis.

In our current version, the structure is organized as follows:

2. Materials and Methods

2.1 Procedure

2.2 Participants

2.3 Instrument

2.4 Data Analysis

We have followed this framework to ensure consistency with previous studies in the field and to maintain clarity and coherence in the presentation of the methodology.

Once again, we appreciate your review and your valuable contributions to improving our manuscript

Comments 5: Practical implications and limitation should be put before conclusions. Conclusions represent the final synthesised narration about the key point of the study, when limitations and implications are taken into account.

Response 5: The suggested modifications have been made, reorganizing the structure of the final section. Now, the practical implications and limitations are presented before the conclusions, ensuring that the synthesized final narrative clearly reflects the key points of the study. Lines 351 to 369 and lines 387 to 404.

Comments 6: Supplementary Material consist of "Revised SWLS" with score interpretation. But this interpretation is not based on the distribution scores of normative population. Therefore, such "norms" can be misleading. Please justify/reconsider.

Response 6: We appreciate your observations regarding the interpretation of scores in the Satisfaction With Life Scale (SWLS) as presented in the supplementary material. After reviewing your comment, we have identified that the inclusion of interpretative score ranges was an oversight.

We recognize that normative scoring studies require probabilistic samples and specific statistical procedures to establish valid reference benchmarks. Since our study does not aim to provide population norms or conduct a score distribution analysis to establish interpretative categories, we have removed these scores from the supplementary material.

We sincerely appreciate your observation, and we have addressed this issue to ensure the methodological rigor of our study.

4. Response to Comments on the Quality of English Language

Point 1:

Response 1: According to the reviewers, the English is correct.

5. Additional clarifications

We sincerely appreciate your valuable feedback. We have incorporated the necessary changes based on the suggestions from both reviewers and also from the second round.

Round 3

Reviewer 1 Report

Comments and Suggestions for Authors

The manuscript has been improved in a satisfactory way. It is advised to implement the information used in the replies to reviewer (especially Comment 2) into the paper. Please clearly indicate that your version of the SWLS uses a 6-point Likert scale. This information is presented in the paper but it should be stated even more clearly. It should be "bold" or very visible as for instance these data (M,SD etc). can be used in meta-analytic studies, when other readers/researchers should clearly know and understand that the response scale is a 6-point. I would suggest mentioning this specific 6-item response scale even in the abstract.

Author Response

For research article

Response to the authors - Third Round

1. Summary

Thank you very much for taking the time to review this manuscript. Please find the detailed responses below and the corresponding revisions.

2. Questions for General Evaluation

Reviewer’s Evaluation

Response and Revisions

Is the content succinctly described and contextualized with respect to previous and present theoretical background and empirical research (if applicable) on the topic?

Can be improved

Thank you very much for your observations; they have been taken into account, applied, and have improved the manuscript.

Are the research design, questions, hypotheses and methods clearly stated?

Can be improved

Thank you very much for your observations; they have been taken into account, applied, and have improved the manuscript.

Are the arguments and discussion of findings coherent, balanced and compelling?

Can be improved

Thank you very much for your observations; they have been taken into account, applied, and have improved the manuscript.

For empirical research, are the results clearly presented?

Can be improved

Thank you very much for your observations; they have been taken into account, applied, and have improved the manuscript.

Is the article adequately referenced?

Yes

. Thank you very much.

Are the conclusions thoroughly supported by the results presented in the article or referenced in secondary literature?

Yes

Thank you very much.

3. Point-by-point response to Comments and Suggestions for Authors

Reviewer 1

Comments 1: The manuscript has been improved in a satisfactory way. It is advised to implement the information used in the replies to reviewer (especially Comment 2) into the paper. Please clearly indicate that your version of the SWLS uses a 6-point Likert scale. This information is presented in the paper but it should be stated even more clearly. It should be "bold" or very visible as for instance these data (M,SD etc). can be used in meta-analytic studies, when other readers/researchers should clearly know and understand that the response scale is a 6-point. I would suggest mentioning this specific 6-item response scale even in the abstract.

Response 1: We appreciate your comments, which have been integrated into the manuscript. The abstract indicates the use of the 6-point version of the SWLS (lines 28 to 29). Likewise, in the Instrument section (2.3, lines 185 to 198), the detailed application of this adapted 6-point scale is described, validated on an Ecuadorian sample according to Schnettler (2017). In addition, the Spanish version of the Satisfaction With Life Scale, with a 6-point scale, has demonstrated adequate validity and reliability in studies conducted in Chile, confirming its unidimensional structure through exploratory and confirmatory factor analyses, as evidenced by previous research by Schnettler et al. (2015, 2017). This underscores the good fit to the data and the stability of its structure in various populations. Finally, lines 377 to 382 suggest continuing to apply this version in other neighboring countries.

4. Response to Comments on the Quality of English Language

Point 1:

Response 1: According to the reviewers, the English is correct.

5. Additional clarifications

We sincerely appreciate your valuable feedback. We have incorporated the necessary changes based on the suggestions from both reviewers and also from the second round.
